# Effects of a Non-Energy-Restricted Ketogenic Diet on Clinical Oral Parameters. An Exploratory Pilot Trial

**DOI:** 10.3390/nu13124229

**Published:** 2021-11-25

**Authors:** Johan Peter Woelber, Christian Tennert, Simon Fabian Ernst, Kirstin Vach, Petra Ratka-Krüger, Hartmut Bertz, Paul Urbain

**Affiliations:** 1Department of Operative Dentistry and Periodontology, Faculty of Medicine, University of Freiburg, Hugstetter Str. 55, 79106 Freiburg, Germany; si.-fa._ernst@web.de (S.F.E.); petra.ratka-krueger@uniklinik-freiburg.de (P.R.-K.); 2Department of Restorative, Preventive and Pediatric Dentistry, University of Berne, Freiburgstrasse 7, 3010 Bern, Switzerland; christian.tennert@zmk.unibe.ch; 3Institute of Medical Biometry and Statistics, Faculty of Medicine, University of Freiburg, Zinkmattenstr. 6A, 79108 Freiburg, Germany; kv@imbi.uni-freiburg.de; 4Department of Medicine I, Medical Center-University of Freiburg, Faculty of Medicine, University of Freiburg, Hugstetter Str. 55, 79106 Freiburg, Germany; hartmut.bertz@uniklinik-freiburg.de (H.B.); paul.urbain@uniklinik-freiburg.de (P.U.)

**Keywords:** ketogenic diet, periodontal inflammation, gingivitis, oral health

## Abstract

Ketogenic diets (KDs) may be a helpful complement in the prevention of and therapy for several diseases. Apart from their non-cariogenic properties, it is still unclear how KDs affect oral parameters. The aim of this study was to investigate the influence of a KD on clinical periodontal parameters. Twenty generally healthy volunteers with an average age of 36.6 years underwent a KD for 6 weeks. Their compliance was monitored by measuring their urinary ketones daily and by keeping 7-day food records. Clinical oral parameters included plaque (PI), gingival inflammation (GI), a complete periodontal status (probing depths, bleeding on probing), and general physical and serologic parameters at baseline and after 6 weeks. The results showed a trend towards lower plaque values, but with no significant changes from baseline to the end of the study with regard to the clinical periodontal parameters. However, their body weight and BMI measurements showed a significant decrease. The regression analyses showed that the fat mass and the BMI were significantly positively correlated to periodontal inflammation, while HDL, fiber, and protein intake were negatively correlated to periodontal inflammation. The KD change did not lead to clinical changes in periodontal parameters in healthy participants under continued oral hygiene, but it did lead to a significant weight loss.

## 1. Introduction

Ketogenic diets (KDs) are characterized by a very low-carbohydrate intake (<10% energy) which induces a state of physiological ketosis with increased use of ketone bodies as an energy source [1]. KDs are used and investigated as potential dietary supports or as therapy for a wide range of diseases, such as intractable childhood epilepsy [2], type 2 diabetes, polycystic ovary syndrome, neurodegenerative diseases [3], and cancer [4,5]. Furthermore, the KD is a popular approach for achieving weight loss [3,6]. While its safety and efficacy were investigated with regard to various anthropomorphic outcomes (body weight, body mass index (BMI), waist circumference, etc.), serological outcomes (cholesterol, markers of blood sugar and insulin resistance, etc.), and body functions (blood pressure, peak oxygen uptake (VO2peak) and peak power, handgrip strength, etc.) [7,8,9], there is a lack of data with regard to clinical oral parameters. This seems surprising since caries are a process that is fundamentally dependent on digested fermentable carbohydrates [10]. Furthermore, there is growing evidence that avoiding sugar and processed carbohydrates in a diet can significantly reduce gingival inflammation, which is a prerequisite for periodontitis, even without changes in plaque values [11,12,13,14]. There are two basic mechanisms assumed about how processed carbohydrates can lead to an alteration of gingival inflammation by a local and a systemic pro-inflammatory effect. Kashket et al. were able to show that the supragingival plaque can metabolize processed carbohydrates to short-chain fatty acids, which, in turn, promote an inflammatory reaction of the gingiva [15]. The systemic effect is assumed to be mediated by high blood sugar peaks with an associated increase in oxidative stress and the formation of advanced glycation end products [10,16]. However, there is also evidence that carbohydrate consumption significantly alters the plaque formation on both teeth and dental implants [17,18], which also has an impact on gingival inflammation [19]. Since caries and periodontitis are the most common diseases in mankind [20], there is a fundamental need for research in therapeutics.

On the other side, there are aspects of KDs which may have a negative impact on periodontal parameters, such as a possible higher intake of saturated fatty acids or an increase in LDL values [16,21]. Furthermore, to the best of the author’s knowledge, it is not described if a drop in serum pH values might have a negative impact on the oral system [22].

Thus, the aim of the current study was to exploratively evaluate the effect of a six-week ketogenic diet on clinical periodontal parameter participants within a larger main study about KDs. Due to the lack of studies regarding KDs and oral parameters, this study focused on generally healthy participants with continued oral hygiene as a first step.

## 2. Materials and Methods

### 2.1. Ethics and Trial Registration

The study was a part of a larger study titled “Impact of a 6-week non-energy-restricted ketogenic diet on physical fitness, body composition and biochemical parameters in healthy adults” [8]. The aim of the main study was to evaluate a six-week KD on endurance and physical performance in young healthy adults because prior studies to this time were only available for athletes. The results showed that KDs had no relevant impact on physical fitness that would impair daily activities or aerobic training [8]. The study protocol was approved by the Ethics Commission of the Albert-Ludwigs-University of Freiburg, Germany (494/14). All subjects signed a written consent form. The study was registered in an international trial register (German Clinical Trial Register number DRKS00009605).

### 2.2. Study Design and Recruitment

This study was designed as an interventional uncontrolled clinical trial. The participants were recruited from employees of the University Medical Center Freiburg via advertising from February to June 2016 at the Department of Medicine I. Besides their participation in the main study, participants were asked to take part in the oral examination conducted at the Department of Operative Dentistry and Periodontology, Faculty of Medicine, University of Freiburg, Germany.

### 2.3. Participants

All participants needed to meet the following inclusion criteria:Adults in good general health with a BMI in the range of 19–30 kg/m^2^;Age ≥ 18 years.

The exclusion criteria were as follows:low-carbohydrate nutrition already prior to the study;Impaired liver and renal function, kidney stones;Pregnancy or lactation period;Diabetes mellitus and any fatty acid-metabolism disorders;Smoking.

### 2.4. Intervention

The experimental intervention consisted of a KD without caloric restriction over a period of 6 weeks in accordance with a modified Atkins diet [23]. Prior to the study period, the participants were instructed by a dietitian. The participants were provided with handouts describing the main aspects of a KD and given a list of suitable foods with a very low carbohydrate content. They were advised to eat ad libitum but to limit their carbohydrate intake to a maximum of 20–40 g/day to derive at least 75%, 15–20%, and 5–10% of total energy from fats, protein, and carbohydrates, respectively. The participants were allowed to vary the food according to their preferences within the framework of the KD. In addition, they were instructed to make the change gradually, step by step, during the first week in order to avoid unwanted side effects (such as constipation) [8]. A detailed description of the foods recommended to avoid and to eat can be found in the supplementary material of Cervenka et al. [24]. Their compliance was monitored by taking daily measurements of urinary ketones using self-testing strips (Ketostix, Bayer Vital GmbH, Leverkusen, Germany) and two semi-quantitative 7-day food records before and during the last week of the intervention. The participants were instructed to measure ketonuria in the morning time since a pre-study with 12 participants measuring both ketonuria and ß-hydroxybutyrate (BHB) in the serum showed that this was the most reliable time for testing [25]. The measurements in the evening and night time showed much higher values for ketonuria compared to serum BHB.

The participants were instructed to accurately record the amounts and types of food and beverages using a digital portable scale (KS 22, Beurer GmbH, Ulm, Germany). The nutrient analysis of the 7-day food records was performed with nutritional database software (Prodi 6.5 basis, Nutri-Science GmbH, Stuttgart, Germany). All participants were advised not to alter their physical activities during the study period. Physical activity was assessed using a validated questionnaire [26].

### 2.5. General and Clinical Oral Measurements

As part of the main study, the fat mass (FM), the body weight, the BMI, and the serological measurements were assessed. The FM was determined via air displacement plethysmography (ADP) using a calibrated BodPod device (Cosmed USA Inc., Concord, CA, USA). The serological measurements included the C-reactive protein (CRP), the blood glucose concentration, insulin, cholesterol, triglycerides (TG), the high-density lipoprotein (HDL), the low-density lipoprotein (LDL), and the insulin-like growth factor (IGF-1).

The clinical oral measurements were assessed by a dentist (SE) and included plaque, gingival inflammation, and a periodontal status including pocket probing depths (PPD), bleeding on probing (BOP), gingival recessions, and the clinical attachment loss (CAL). Based on PPDs and BOP, the periodontal inflamed surface area (PISA) was calculated [27]. Furthermore, participants were interviewed with regard to their oral hygiene behavior, including oral hygiene products and the frequency of use. Plaque was measured with the plaque index (PI) described by Silness and Löe [28] using a dental probe (Emil Huber GmbH, Karlsruhe, Germany). Gingival inflammation (GI) was measured according to Löe and Silness [28]. Periodontal measurements were performed with a pressure-sensitive periodontal probe DB764R (Aesculap AG, Tuttlingen, Germany). In order to assess the intra-rater reliability regarding the PPDs, half of the participants were measured twice in the first quadrant after the assessment of the periodontal status.

### 2.6. Statistical Analysis

The sample size calculation was carried out within the main study for the primary outcome peak oxygen uptake (VO2peak) [8]. Accordingly, the current study had no predefined primary and secondary outcomes but followed an explorative approach. The median, mean, standard deviation, minimum and maximum were calculated for a descriptive description of the data. A graphical representation was made by boxplots. By means of t-tests for paired samples, a before–after comparison was performed. A linear regression model was fitted per time point for an investigation of the influence of different influencing variables on the clinical parameters. It was also used to examine the changes in the clinical parameters. Due to the explorative character of this study, no correction for multiple testing was carried out. All data were analyzed using STATA 14.1 (StataCorp LP, Lakeway Drive College Station, TX, USA). *p* < 0.05 was considered as statistically significant.

## 3. Results

Twenty participants were recruited and completed the investigation, without any dropouts. The group consisted of 16 women and 4 men with a mean age of 36.6 years varying from 25 to 57 years. With regard to oral hygiene procedures, all participants brushed their teeth between 1 and 3 times daily and used additional oral hygiene products in varying frequencies (Table 1).

### 3.1. Intragroup Differences

The factors PI, GI, BOP, PPD, CAL, and PISA did not change significantly from baseline to the end. With regard to PI, there was a trend toward lower scores at the final examination (*p* = 0.096; Table 2). The repeated measurements of the first quadrant showed high intra-rater reliability with an intraclass-correlation (ICC) of 0.98.

**Table 2 nutrients-13-04229-t002:** Intragroup differences for clinical periodontal parameters from baseline to the final examination.

Oral Parameter	Time	Mean (Standard Deviation)	(95%Conf. Interval)	Intra- *p*-Value (Baseline vs. End)
PI	Baseline	0.49 (0.33)	0.34–0.65	0.0969
	End	0.42 (0.27)	0.29–0.55	
GI	Baseline	0.68 (0.28)	0.55–0.82	0.1814
	End	0.62 (0.23)	0.51–0.73	
BOP	Baseline	0.29 (0.09)	0.24–0.33	0.2601
	End	0.27 (0.11)	0.22–0.32	
PPD	Baseline	2.25 (0.22)	2.15–2.36	0.2743
	End	2.19 (0.17)	2.11–2.27	
CAL	Baseline	2.59 (0.41)	2.40–2.78	0.8156
	End	2.60 (0.40)	2.41–2.79	
PISA	Baseline	370.1 (141.5)	303.9–436.4	0.2545
	End	341.3 (149.0)	271.5–411.1	

PI = plaque index; GI = gingival index; BOP = bleeding on probing; PPD = pocket probing depth; CAL = clinical attachment level; PISA = periodontal inflamed surface area. With regard to the anthropomorphic data and physical activity, body weight and BMI showed a significant decrease from baseline to the end (Table 3). Regarding the fat mass, there was a trend towards lower values at the final examination.

**Table 3 nutrients-13-04229-t003:** Intragroup differences for anthropomorphic parameters and physical activity from baseline to the final examination.

Parameter	Time	Mean (Standard Deviation)	(95% Conf. Interval)	Intra- *p*-Value (Baseline vs. End)
Body weight (kg)	Baseline	69.20 (10.65)	64.21–74.17	0.0003 **
	End	67.34 (9.26)	63.01–71.67	
BMI (kg/m^2^)	Baseline	23.60 (3.25)	22.08–25.13	0.0002 **
	End	22.97 (2.77)	21.67–24.26	
Fat mass (kg)	Baseline	22.75 (9.32)	18.39–27.11	0.0675
	End	21.99 (8.33)	18.09–25.89	
Physical activity (MET-hours/week)	Baseline	39.31 (19.51)	30.17–48.44	0.1600
	End	43.29 (24.47)	31.83–54.74	

MET = metabolic equivalent of task., ** *p* < 0.01.

The serological parameters, glucose, and IGF-1 showed a significant decrease from baseline to the end (Table 4). The LDL value showed a significant increase from baseline to the end, while the HDL did not change from baseline to the end. Changes in nutrient intake from baseline to the final examination are presented in Table 5.

### 3.2. Regression Analysis

Regarding the age, gender, and oral hygiene behavior parameters (interdental cleaning and use of oral mouthwashes), there were no significant associations to clinical parameters neither at baseline nor in the end. The results of the regression analysis with regard to anthropomorphic parameters, physical activity, serological parameters, and dietary parameters are shown in Table 6. With regard to the clinical inflammatory parameter (GI, BOP), there were significant pro-inflammatory associations to the BMI and fat mass and significantly anti-inflammatory associations to the HDL, fiber, and protein intake. With regard to plaque, there were also significantly positive associations to the BMI and fat mass and significantly negative associations to the MET, HDL, Kcal, and fat and protein intake.

## 4. Discussion

### 4.1. Changes in Clinical Oral Parameters

The aim of the present study was to exploratively investigate the effects of a very low-carb, ketogenic diet on clinical oral parameters along with serological, anthropomorphic, and nutritional parameters in an uncontrolled clinical trial. The results showed that after six weeks of intervention, there were no significant changes in the clinical oral parameters, except a trend towards lower plaque values. Accordingly, the ketogenic diet did not lead to a worsening nor an increase in periodontal health. This result must be discussed with regard to various factors. First, it must be mentioned that the investigated population consisted of healthy, relatively young volunteers with adequate oral hygiene and low baseline plaque and gingivitis values. The mean GI value of 0.68 represented a value between 0 (=no gingivitis) to 1 (=mild gingivitis), with no participants showing severe gingivitis [28]. Compared to an epidemiological study by Li et al. showing a mean GI-value of 1.06 in a population of 1000 US participants with a comparable mean age of 37.9 years, the investigated population showed less gingivitis. This might be due to continued oral hygiene measures since oral hygiene is very effective in gingivitis reduction [21]. This is in accordance with the presented low mean PI of 0.49, which is in a comparable low range of young, healthy adults evaluated in a German population [29]. With regard to the found trend towards lower plaque values at the end of the study, an additional post hoc power analysis showed that with 20 patients, a difference in PI of 0.2 could have been detected with 80% power and an alpha of 5%.

Against the background of the rather low GI and PI values, the current study setting was both limited to allow a further deterioration due to the high efforts in oral hygiene as well as to show further positive effects on GI since the baseline values were in a low range already. This might be very different to an obese, periodontitis- or gingivitis-affected population. Considering this together, further investigations of ketogenic diets on participants with higher plaque values would be interesting. Baumgartner et al. showed that even under the total absence of oral hygiene for four weeks, participants did experience a stable low level of gingivitis (G = ~0.4) and even a significant decrease in BOP (Δ−12.2%) during a sugar-free stone-age-adapted diet [14]. This is in accordance with a meta-analysis of Hujoel showing a profound gingivitis-decreasing effect when reducing the dietary sugar [11] and previous investigations of our own working group looking at low-carb diets and related gingival effects [12,13]. These studies indicate that the classically assumed correlation between plaque and gingival inflammation might only apply to pro-inflammatory Western diet conditions with high consumption of processed carbohydrates, sugar, saturated fats and animal proteins.

With regard to the study duration and the anticipated quick response of the gingival tissues based on previous findings (within even 4 weeks of intervention), the current study duration of 6 weeks would have been able to show clinical changes [12,13,14].

Additionally, from a cariogenic point of view, it would be interesting to study the effects of ketogenic diets on the oral microbiome since there is a tremendous lack of interventional studies on sugar-free (or even very low-carb diets) and caries development and/or progression. Behind the background of current theories on caries etiology, the almost total absence of fermentable carbohydrates would not allow any cariogenic shifts in the oral microbiome or demineralization of dental hard tissues [10]. To the best knowledge of the authors, there are only two interventional studies looking at the effects of a sugar-free diet on oral microbiota [30,31]. Both studies showed a significant reduction of the cariogenic strain *Streptococcus mutans* due to the omitted sugar. Translating these results into the practical applications of a ketogenic diet, patients might also benefit from caries protection. This is highly relevant for cancer patients since they are often compromised with xerostomia due to radiation or immunotherapy [32].

### 4.2. Changes in Non-Oral Parameters

With regard to the anthropomorphic parameters, there was a significant decrease in body weight and BMI. This result is in line with several investigations showing a weight-reducing effect of ketogenic diets [3,6], which is hypothesized to be due to higher rates of resting energy expenditure, lower levels of insulin, and higher satiety [33,34]. Compared to the general German population, this mean weight loss of about 1.9 kg happened in participants with a normal BMI, whereas more than half of the age-matched general population suffers from overweight or obesity [35].

Even if the long-term suitability of ketonic diets has to be questioned, periodontitis patients who are overweight might benefit from the recommendation to at least reduce the intake of processed carbohydrates since being overweight is also a dominant risk factor for periodontal inflammation [36]. Interestingly, in this study, BMI also showed several significant correlations to oral inflammatory parameters such as GI, BOP, and PISA. However, the current ketogenic diet intervention was also accompanied by an increase in serological LDL and total cholesterol levels and an increased intake of saturated fats and dietary cholesterol. This has to be discussed critically since these parameters are not only associated with higher periodontal inflammation but also with general diseases, such as cardiovascular diseases [37,38], and the increase in LDL cholesterol seems to be common in ketogenic diets [9,33]. However, compared to a general Western population, the observed LDL values were still in a low range [38].

With regard to the possible anti-inflammatory properties of omega-3 fatty acids on periodontal inflammation [39], the study could not show any associations. This could be due to the generally low clinical effect previously discussed or to the simultaneous increase in omega-3 and omega-6 fatty acids, whose balance is shown to be an important factor for the clinical effects of omega-3 fatty acids [40]. This factor could be improved in future studies to recommend the intake of mainly plant-derived fats and proteins instead of animal products, which—also within a low-carb diet—would be beneficial to overall health [41]. The recommendations of the current intervention included both animal- and plant-based products and foods. This further differentiation between the sources of macronutrients would also be very important with regard to the effects on the gut microbiome due to the marked difference between the included pre- and probiotics [42].

### 4.3. Limitations

The study was clearly limited by a low number of participants and a missing control group, which could verify the association of the observed effects to the intervention. However, with regards to the clinical oral effects, it would be unlikely to observe different outcomes under continued oral hygiene measures. Another aspect is that the intervention and control groups can hardly be blinded, which, however, also applies to other dietary studies. On the other side, the intake of non-blinded foods is the condition of real life.

The anthropomorphic effects were all in line with already described effects in the literature. With regard to the serum values, an additional continuous measurement of serum ketosis would be recommended in future studies to ensure the state of ketosis. This should also include pH measurements of the serum and the saliva.

Furthermore, an investigation of the oral microbiome would have allowed a further interpretation of caries- and periodontitis-associated microbiota. This should be included in further studies. Due to the unequal distribution of women and men, future studies might also stratify with regard to gender.

## 5. Conclusions

The ketogenic dietary change did not lead to clinical changes in periodontal parameters in healthy participants under continued oral hygiene, but it did lead to significant weight loss.

## Figures and Tables

**Table 1 nutrients-13-04229-t001:** Oral hygiene behavior of the experimental group.

Oral Hygiene Behavior	Mean Frequency (Per Day)	Standard Deviation	Minimum–Maximum
Tooth brushing	1.95	0.39	1–3
Interdental cleaning	0.49	0.55	0–2
Use of mouth rinse	0.41	0.68	0–2

**Table 4 nutrients-13-04229-t004:** Intragroup differences for serological parameters from baseline to the final examination.

Serological Parameter	Time	Mean (Standard Deviation)	(95% Conf. Interval)	Intra- *p*-Value (Baseline vs. End)
CRP (mg/L)	Baseline	1.75 (2.31)	0.67–2.83	0.2127
	End	3.05 (5.00)	0.71–5.39	
Glucose (mg/dL)	Baseline	90.55 (6.66)	87.43–93.67	0.0436 *
	End	87.75 (5.56)	85.15–90.35	
IGF-1 (ng/mL)	Baseline	197.15 (86.10)	156.85–237.45	0.0055 **
	End	154.50 (66.83)	123.22–185.78	
Insulin (pmol/L)	Baseline	53.00 (27.77)	40.00–66.00	0.2259
	End	45.50 (20.12)	36.08–54.92	
HDL (mg/dL)	Baseline	75.65 15.11)	68.58–82.72	0.6235
	End	76.65 (16.65)	68.86–84.44	
LDL (mg/dL)	Baseline	107.80 (38.00)	90.01–125.59	0.0495 *
	End	118.40 (37.51)	100.85–135.95	
Total cholesterol (mg/dL)	Baseline	187.70 (40.59)	168.70–206.70	0.3283
	End	193.05 (37.02)	175.72–210.38	
Triglycerides (mg/dL)	Baseline	76.70 (45.96)	55.19–98.21	0.3505
	End	68.10 (29.52)	54.29–81.91	

CRP = C-reactive Protein, IGF-1 = insulin-like growth factor 1, HDL = high-density lipoprotein, LDL = low-density lipoprotein. * *p* < 0.05, ** *p* < 0.01.

**Table 5 nutrients-13-04229-t005:** Intragroup differences for nutrient intake from baseline to the final examination.

Nutritional Parameter	Time	Mean (Standard Deviation)	(95% Conf. Interval)	Intra- *p*-Value (Baseline vs. End)
Kcal	Baseline	2245.03 (371.00)	2071.40–2418.66	0.7576
	End	2209.53 (561.18)	1956.89–2472.17	
Proteins (g)	Baseline	77.68 (13.15)	71.52–83.83	<0.0001 **
	End	104.54 (20.77)	94.82–114.26	
Fat (g)	Baseline	96.39 (28.23)	83.18–109.61	<0.0001 **
	End	173.38 (56.20)	147.08–199.68	
Carbohydrates (g)	Baseline	230.93 (46.43)	209.20–252.66	<0.0001 **
	End	42.10 (13.18)	35.93–48.27	
Fibre (g)	Baseline	23.72 (6.02)	20.91–26.54	0.9497
	End	23.62 (7.07)	20.31–26.92	
Cholesterol (mg)	Baseline	318.69 (132.90)	256.49–380.89	<0.0001 **
	End	457.04 (120.86)	400.47–513.60	
Saturated fats (g)	Baseline	41.18 (12.11)	35.51–46.85	<0.0001 **
	End	67.30 (25.10)	55.55–79.04	
Monounsaturated fats (g)	Baseline	31.84 (11.12)	26.63–37.04	<0.0001 **
	End	63.01 (22.47)	52.49–73.52	
Polyunsaturated fats (g)	Baseline	13.38 (5.72)	10.71–16.06	<0.0001 **
	End	24.20 (8.69)	20.14–28.27	
Linolenic acid (n-3) (g)	Baseline	1.91 (1.56)	1.18–2.64	0.0036 **
	End	3.71 (2.78)	2.41–5.01	
Linoleic acid (n-6) (g)	Baseline	10.90 (4.44)	8.82–12.98	<0.0001 **
	End	17.44 (7.34)	14.00–20.88	
Eicosapentaenoic acids (EPA) (g)	Baseline	0.05 (0.08)	0.01–0.08	0.0055 **
	End	0.17 (0.15)	0.10–0.24	
Docosahexaenoic acids (DHA) (g)	Baseline	0.13 (0.15)	0.06–0.20	0.0204 *
	End	0.29 (0.24)	0.18–0.40	

* *p* < 0.05, ** *p* < 0.01.

**Table 6 nutrients-13-04229-t006:** Regression coefficients of the linear regression of different parameters on clinical oral parameters at baseline (T1) and at the end of the study period (T2).

	Clinical Oral Parameter									
	PI		GI		PPD		BOP		PISA	
Parameter	T1	T2	T1	T2	T1	T2	T1	T2	T1	T2
BMI	0.053 *	0.064 *	0.052 *	0.032	0.005	0.033 *	0.014 *	0.011	22.323 *	21.807
MET	−0.010 **	−0.006 *	−0.005	−0.004	−0.004	−0.002	−0.001	−0.001	−2.282	−2.472
Fat mass	0.012	0.019 *	0.013	0.006	0.001	0.010 *	0.005 *	0.005	6.902 *	8.312 *
Glucose	−0.014	−0.001	−0.007	−0.008	−0.009	0.001	0.004	0.002	4.451	4.909
CRP	0.009	0.015	0.034	0.014	−0.008	−0.001	0.013	0.002	16.121	2.636
HDL	−0.009	−0.007 *	−0.007	−0.007 *	−0.001	−0.004	−0.001	−0.002	−1.056	−2.867
LDL	0.004	0.003	0.002	0.001	0.001	0.001	0.001	−0.001	1.378	−0.841
Kcal	0.001	−0.001 *	0.001	−0.001	−0.001	−0.001 *	−0.001	−0.001	−0.106	−0.113
CHOs	−0.001	0.001	−0.001	0.001	−0.001	0.001	−0.001	0.001	−1.393 *	1.399
Fat	0.001	−0.003 *	0.001	−0.001	−0.001	−0.002 *	−0.001	−0.001	−0.409	−1.062
Proteins	0.001	−0.007 *	0.005	−0.003	−0.005	−0.005 **	0.001	−0.003 *	−0.081	−3.860 *
Fibre	−0.020	−0.016	−0.014	−0.018 *	−0.005	−0.002	−0.003	−0.005	−7.768	−6.899
EPA	−0.741	0.366	0.237	0.660	0.577	−0.187	0.253	0.102	652.12	91.365
DHA	−0.523	0.247	0.006	0.383	0.224	−0.107	0.138	0.077	358.164	88.329

PI = plaque index; GI = gingival index; PPD = pocket probing depth; BOP = bleeding on probing; PISA = periodontal inflamed surface area; CRP = C-reactive Protein; HDL = high-density lipoprotein, LDL = low-density lipoprotein; BMI = body mass index; MET = metabolic equivalent of task; CHOs = carbohydrates; EPA = Eicosapentaenoic acids; DHA = Docosahexaenoic acids; * *p* < 0.05, ** *p* < 0.01.

## Data Availability

The datasets used and/or analyzed during the current study are available from the corresponding author on reasonable request.

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
