# Peer review of "Effects of a Non-Energy-Restricted Ketogenic Diet on Clinical Oral Parameters. An Exploratory Pilot Trial"

_nutrients, 2021, doi:10.3390/nu13124229_

Round 1
Reviewer 1 Report
Introduction
Periodontal disease caused by bacteria. Why you think that ketogenic influence the periodontal parameter?
Methods
In line 96, please explain add father information in manuscripts or supplemental data.
How did you investigate the oral health behavior?
Who did check the oral parameters?
Please add the significant value of P in statistical analysis section.
Results
You should show the number of oral health behavior (brushing, interdental brush, and dental rinse) .
Discussion
Please mention participants were not special by comparing to general people in oral, anthropomorphic, and serological parameters.
Author Response
Thank you for your time and efforts to support the quality of our manuscript. Please find in the following your comments accompanied with our corresponding answers.
Introduction
- Periodontal disease caused by bacteria. Why you think that ketogenic influence the periodontal parameter?
Thank you for your comment. We added further information to the introduction now.
Added content (l53): “There are two basic mechanisms assumed, how processed carbohydrates can lead to an alteration of gingival inflammation, by a local and a systemic pro-inflammatory effect. Kashket et al. were able to show that the supragingival plaque can metabolize processed carbohydrates to short-chain fatty acids which in turn promote an inflammatory reaction of the gingiva [15]. The systemic effect is assumed to be mediated by high blood-sugar peaks with an associated increase in oxidative stress and the formation of advanced glycation end products [10,16].”
Methods
- In line 96, please explain add father information in manuscripts or supplemental data.
Thank you. We added more information about the dietary instructions.
Added content (l115): “The participants were allowed to vary the food according to their preferences within the framework of the KD. In addition, they were instructed to make the change gradually step by step during the first week in order to avoid unwanted side effects (such as constipation) [8]. A detailed description of foods recommended to avoid and to eat can be found in the supplementary material of Cervenka et al. [22].”
- How did you investigate the oral health behavior?
The oral health behavior was assessed by verbal anamnesis, described in the section “2.5. General and clinical oral measurements”. We added more details now.
Added content (l147): “Furthermore, participants were interviewed with regard to their oral hygiene behavior including oral hygiene products and the frequency of use.”; + Table 1
- Who did check the oral parameters?
Oral parameters were assessed by one dentist (SE). We added this information to the text now.
Added content (l142): “Clinical oral measurements were assessed by a dentist (SE) and included plaque, gingival inflammation …”
- Please add the significant value of P in statistical analysis section.
Comment to the reviewer: Thank you for this suggestion. We added the value to the text now.
Added content (l169): “p<0.05 was considered as statistically significant.”
Results
- You should show the number of oral health behavior (brushing, interdental brush, and dental rinse)
Thank you for your comment. Additionally to a comment by reviewer #2, we changed the content of table 1 from the demographic data to the oral hygiene behavior.
Added content: Table 1
Discussion
- Please mention participants were not special by comparing to general people in oral, anthropomorphic, and serological parameters.
Thank you for this suggestion. We added a comparison to the main variables now.
Added content (l238): “Compared to an epidemiological study by Li et al. showing a mean GI-value of 1.06 in a population of 1000 US-participants with a comparable mean age of 37.9 years, the investigated population showed less gingivitis. This might be due to continued oral hygiene measures since oral hygiene is very effective in gingivitis reduction [21]. This is in accordance with the presented low mean PI of 0.49 which is in a comparable low range of young healthy adults evaluated in a German population [27].”
L294: ”Compared to the general German population this mean weight loss of about 1.9 kg happened in participants with a normal BMI whereas more of half of the age-matched general population suffers from overweight or obesity [33].”
L310: “However, compared to a general western population the observed LDL-values were still in a low range [36].”
Reviewer 2 Report
The topic of ketogenic diets is timely given the widespread interest in using the diet as a therapeutic and also as a ‘popular’ diet aimed at achieving rapid weight loss. While I recognize that these data were collected as part of a previous study, it is important to provide justification for the study design details and reporting of outcomes included in this submission. This submission makes an interesting addition to the existing literature in the field of nutrition and oral health.
Major comments:
Please justify why a 6-week intervention was used. Based on previous findings of this research team and others, would it be anticipated that this duration of study would result in differences in clinical outcomes? Discussion of this aspect should be included in the submission.
This point leads to another question - what is the primary outcome for this secondary analysis? Please clearly state the primary outcome along with the corresponding rationale in the submission.
The rational for why it was anticipated that the ketogenic diet would have a negative effect on clinical oral parameters requires clarification. For example, line 204 mentions this aspect though I wondered why a negative effect is anticipated? (this seems to be misaligned with text in lines 47-52)
Further, a rational for studying a healthy population with correspondingly healthy oral health parameters needs to be included, preferably in the Introduction.
Given that many countries are increasingly encouraging greater intakes of plant-based proteins, including additional information about type of protein (animal versus plant sources) would be helpful and timely. Also, discussion around intakes of dairy and non-dairy protein would be useful to ongoing discussions surrounding different types of animal protein, particularly as some dairy foods include fermented foods.
Also including information about type of fiber, and specifically prebiotics, would add to the dietary information and provide useful information for researchers designing future studies.
Comments about statistical analysis:
State what the main outcome was for the initial sample size calculation (Line 129). I suggest that a post-hoc sample size or power analysis be performed for the present study based on the primary outcome for the present study.
Was there a correction used to account for the many statistical comparisons being made? Is Table 5 too exploratory?
Was sex considered within the model and/or analyses? Should there be a statement about this aspect?
Data from Table 1 can be included as text in the Results rather than as a separate table.
Minor revisions:
First sentence of the abstract (Line 16). It wasn’t clear why only oncological therapy is mentioned when this diet has a long history of use for other conditions. As is, the statement seems out of place. I suggest removing the reference to oncological therapy and including a more general introductory statement.
Line 40. Consider removing ‘eventually’.
Provide additional details about the original study and why healthy, young adults were studied – what was the main outcome of the original study? Why was smoking an exclusion criteria? How would excluding smokers have potentially altered the study findings?
Include the error of measurement of urinary ketones using self-testing strips.
It would be very interesting to include some examples of major foods that were eliminated and added to the diet during the study. I mention this as I feel it would add a helpful practical aspect to the study. Could include some sample diets of individuals over a 24 hour period as a supplemental file.
Line 209-210. I wondered whether ethics could be obtained for the study mentioned in which oral hygiene is deliberately compromised given the links between systemic and oral health. Perhaps remove this statement as an idea for a future study.
Discussion about why there is no control group would add to the submission. Diet studies are so difficult to ‘blind’ and I think that the design used is appropriate. Highlighting the practical aspects of diet intervention studies may be useful and add to the ongoing discussion about this aspect among researchers that study the effects of dietary changes on health outcomes.
Author Response
Dear Reviewer, thank you for your time and efforts to support the quality of our manuscript. Please find following your comments accompanied with our corresponding answers.
Major comments:
- Please justify why a 6-week intervention was used. Based on previous findings of this research team and others, would it be anticipated that this duration of study would result in differences in clinical outcomes? Discussion of this aspect should be included in the submission.
Thank you for your comment. The duration of 6 weeks was mainly determined by the main study. However, based on previous findings of our research team and others even shorter durations of 4 weeks of dietary changes are adequate to see clinical changes (Woelber et al. 2016/2019, Baumgartner et al. 2009). We added this aspect to the discussion part now. It is really interesting that the gingiva seems to respond that quickly to dietary changes.
Added content (l270): “With regard to the study duration and the anticipated quick response of the gingival tissues based on previous finding (within even 4 weeks of intervention) the current study duration of 6 weeks would have been able to show clinical changes [12–14].”
- This point leads to another question - what is the primary outcome for this secondary analysis? Please clearly state the primary outcome along with the corresponding rationale in the submission.
Thank you for this point. Due to the fact that the sample size calculation was performed for the primary outcome parameter “peak oxygen uptake (VO2peak)” within the main study (Urbain et al. 2017), our substudy had an explorative approach. Accordingly, we were not able to formulate primary and secondary outcomes. We emphasize this circumstances in the introduction and material/methods part now. Furthermore, we added a post hoc power calculation for the PI now.
Added content (l157): “The sample size calculation was carried out for within the main study for the primary outcome peak oxygen uptake (VO2peak) [8]. Accordingly, the current study had no predefined primary and secondary outcomes, but followed an explorative approach.”
+L245: “With regard to the found trend towards lower plaque values at the end of the study, an additional post-hoc power analysis showed that with 20 patients a difference in PI of 0.2 could have been detected with 80% power and an alpha of 5%.”
- The rational for why it was anticipated that the ketogenic diet would have a negative effect on clinical oral parameters requires clarification. For example, line 204 mentions this aspect though I wondered why a negative effect is anticipated? (this seems to be misaligned with text in lines 47-52)
Thank you for this clarification. We added an explanation in the introduction now.
Added content (l66): “On the other side, there are aspects of KD which may have a negative impact on periodontal parameters like a possible higher intake of saturated fatty acids or an increase in LDL-values [16,21].”
- Further, a rational for studying a healthy population with correspondingly healthy oral health parameters needs to be included, preferably in the Introduction.
Thank you for this suggestion. Due to possible unknown negative effects (see point 3) we added this information to the introduction now.
Added content (l69): “Thus, the aim of the current study was to exploratively evaluate the effect of a six-week ketogenic diet on clinical periodontal parameters of general healthy participants within a larger main study about KD. Due to the lack of studies regarding KD and oral parameters this study focused on general healthy participants with continued oral hygiene in a first step.”
- Given that many countries are increasingly encouraging greater intakes of plant-based proteins, including additional information about type of protein (animal versus plant sources) would be helpful and timely. Also, discussion around intakes of dairy and non-dairy protein would be useful to ongoing discussions surrounding different types of animal protein, particularly as some dairy foods include fermented foods.
Thank you for this specification. Due to personal changes in our partner institution and the used nutritional software we´re not able to deliver further results differentiating between animal and plant-derived proteins. However, we emphasize the dietary recommendations in the manuscript now, which included both animal and plant products. This applies also for the intake of dairy and fermented foods.
Added content (l317): “This factor could be improved in future studies to recommend the intake of mainly plant-derived fats and proteins instead of animal products which – also within a low-carb diet – would be beneficial to overall health [40]. The recommendations of the current intervention included both animal- and plant-based products and foods. This further differentiation between the sources of macronutrients would also be very important with regard to the effects on the gut microbiome due to the marked difference between included pre- and probiotics [41].”
- Also including information about type of fiber, and specifically prebiotics, would add to the dietary information and provide useful information for researchers designing future studies.
Thank you for this point. In accordance to point 5, we added this point to the text now.
Added content (l322): “This further differentiation between the sources of macronutrients would also be very important with regard to the effects on the gut microbiome due to the marked difference between included pre- and probiotics [41].”
Comments about statistical analysis:
- State what the main outcome was for the initial sample size calculation (Line 129). I suggest that a post-hoc sample size or power analysis be performed for the present study based on the primary outcome for the present study.
Thank you for this point. We added ad post-hoc power analysis for the PI in the text now (see also point 2).
Added content (l245): “With regard to the found trend towards lower plaque values at the end of the study, an additional post-hoc power analysis showed that with 20 patients a difference in PI of 0.2 could have been detected with 80% power and an alpha of 5%.”
- Was there a correction used to account for the many statistical comparisons being made? Is Table 5 too exploratory?
As mentioned in the statistical methods part now no correction for multiple testing was done due to the explorative character of this study.
- Was sex considered within the model and/or analyses? Should there be a statement about this aspect? LIMITATIONS
Thank you. Due to the very unequal gender distribution (4 male, 16 female), sex was not included as a covariate. However, our analyses in advance did not show any indication of gender dependence of the parameters. We also added this aspect in the limitations part.
Added content (l336): “Due to the unequal distribution of women and men future studies might also stratify with regard to gender.”
- Data from Table 1 can be included as text in the Results rather than as a separate table.
Done. According to Reviewer #1 we present the oral hygiene behaviors in table 1 now.
Minor revisions:
- First sentence of the abstract (Line 16). It wasn’t clear why only oncological therapy is mentioned when this diet has a long history of use for other conditions. As is, the statement seems out of place. I suggest removing the reference to oncological therapy and including a more general introductory statement.
Thank you for this comment. We changed this first sentence accordingly.
- Line 40. Consider removing ‘eventually’.
Done.
- Provide additional details about the original study and why healthy, young adults were studied – what was the main outcome of the original study?
Thank you for this point. We added more information about the main study now.
Added content (l69): “Thus, the aim of the current study was to exploratively evaluate the effect of a six-week ketogenic diet on clinical periodontal parameters of general healthy participants within a larger main study about KD. Due to the lack of studies regarding KD and oral parameters this study focused on general healthy participants with continued oral hygiene in a first step.”
+L157: “The sample size calculation was carried out within the main study for the primary outcome peak oxygen uptake (VO2peak) [8].”
- Why was smoking an exclusion criteria? How would excluding smokers have potentially altered the study findings?
Thank you for this specification. This exclusion criterion was indeed determined by our substudy, because smoking has such a profound impact on gingival bleeding due to vasoconstriction. Accordingly, smoking would have masked possible signs of gingivitis. Please find more information here: Chapple et al. Periodontal health and gingival diseases and conditions on an intact and a reduced periodontium: Consensus report of workgroup 1 of the 2017 World Workshop on the Classification of Periodontal and Peri-Implant Diseases and Conditions. J Periodontol. 2018 Jun;89 Suppl 1:S74-S84.
- Include the error of measurement of urinary ketones using self-testing strips.
Thank you for this comment. Due to personal changes in our partner institution we were not able to include the errors of measurement. However, we added further information about the validity of the used measurement due to a prestudy which investigated the best time for testing the ketonuria: Urbain P, Bertz H. Monitoring for compliance with a ketogenic diet: what is the best time of day to test for urinary ketosis? Nutr Metab (Lond). 2016 Nov 4;13:77. doi: 10.1186/s12986-016-0136-4. PMID: 27822291
Added content (l124): “The participants were instructed to measure ketonuria in the morning time, since a prestudy showed that this was the most reliable time for testing [23].”
- It would be very interesting to include some examples of major foods that were eliminated and added to the diet during the study. I mention this as I feel it would add a helpful practical aspect to the study. Could include some sample diets of individuals over a 24 hour period as a supplemental file.
Thank you for this suggestion, which we totally agree on. We added ad reference to the detailed reccomendations for the participants now. According to point 15, we are not able to present an example of the food diary.
Added content (l119): “A detailed description of foods recommended to avoid and to eat can be found in the supplementary material of Cervenka et al. [22].”
- Line 209-210. I wondered whether ethics could be obtained for the study mentioned in which oral hygiene is deliberately compromised given the links between systemic and oral health. Perhaps remove this statement as an idea for a future study.
Thank you for this question. We are sure that due to the amount of studies, which show that oral health is possible under the absence of oral hygiene, the ethics would be obtainable. Besides, why should homo sapiens be the only species who is dependent on oral hygiene. However, we understand your thoughts and have softened the intense of this sentence.
Changed content (l258): “Taken this together, further investigations of ketogenic diets on participants with higher plaque values would be interesting.”
- Discussion about why there is no control group would add to the submission. Diet studies are so difficult to ‘blind’ and I think that the design used is appropriate. Highlighting the practical aspects of diet intervention studies may be useful and add to the ongoing discussion about this aspect among researchers that study the effects of dietary changes on health outcomes.
Thank you for this interesting point. We added a discussion about the ´blinding´ in the limitations part now.
Added content (l330): “Another aspect is that the intervention and control groups can hardly be blinded, which however applies also for other dietary studies. On the other side, the intake of non-blinded foods is the condition of real life.”
Round 2
Reviewer 1 Report
Thank you for your revising of this manuscripts.
I feel enough to publish in "Nutrients".
Author Response
Thank you